# Metabolome Analysis of the Effects of Sake Lees on Adipocyte Differentiation and Lipid Accumulation

**Yuki Motono [1,*], Shin Nishiumi [2,3], Masaru Yoshida [2,4,5] and Motoko Takaoka [3]**

[1] Department of Human Science, Graduate School, Kobe College, Nishinomiya 662-0825, Japan
[2] Division of Gastroenterology, Department of Internal Medicine, Kobe University Graduate School of Medicine, Kobe 650-0017, Japan
[3] Department of Biosphere Sciences, Kobe College, Nishinomiya 662-0825, Japan
[4] Research Institute of Food and Nutritional Sciences and Department of Food Science & Nutrition, School of Human Science & Environment, University of Hyogo, Himeji 670-0092, Japan
[5] Metabolomics Research, Department of Internal Related, Kobe University Graduate School of Medicine, Kobe 650-0017, Japan
[*] Correspondence: yu.t1vx3q.ki@gmail.com; Tel.: +81-798-51-8423

**Abstract:** Obesity, along with hypertension and hyperlipidemia, is one of the leading factors of metabolic syndrome, which increases the risk of diabetes. However, controlling obesity is a global challenge. Sake lees, or Japanese rice wine lees, is a by-product of sake fermentation and has been consumed in Japan for a long time. Sake lees contains an abundance of amino acids, peptides, dietary fiber, and micronutrients, which make it highly nutritional. Additionally, sake lees has been reported to have multiple interesting effects when ingested and may aid in combating obesity. In this study, we investigated the effects of sake lees materials on preadipocyte differentiation and fat accumulation in preadipocyte cells (3T3-L1) and analyzed it with a metabolome analysis. We found that compared to the control group, lipid accumulation was suppressed by 80.9% when the 100 °C extract of indigestible sake lees component (ISLCs) was added to 1 mg/mL. Additionally, the metabolome analysis revealed various other differences between the control group and the group treated with ISLCs, especially in amino acids concentrations. Based on the above findings, we demonstrate that ISLCs affect the amino acid metabolic pathways, which in turn affect differentiation and lipid accumulation in adipocytes. Therefore, we suggest that sake lees may aid in combating obesity and addressing metabolic syndromes, both of which can be considered as global issues. The limitation of this research is sake lee is a general non-direct edible raw material and it is difficult to add as a regular diet.

**Keywords:** sake lees; adipocyte; lipid accumulation; metabolome analysis





## 1. Introduction

In the 1980s, the mean lifespan of the Japanese general population became the world's longest [1]. While Japan is still known as a country with a healthy and long-living populace, the proportion of obese people has recently increased because of changes in lifestyle such as the adoption of Westernized diets [2,3]. The prevalence of obesity in Japan is now a social issue [4].

Obesity results from an accumulation of excess triglycerides stored in adipocyte cells when energy intake exceeds energy expenditure. Adipocytes usually secrete hundreds of adipocytokines that have proinflammatory and anti-inflammatory properties. When adipocytes increase in size due to excessive energy intake, they produce more proinflammatory adipocytokines. The presence of malignant adipocytokines is closely related to the development of lifestyle-related diseases; therefore, adipocyte proliferation is linked to the occurrence of lifestyle-related diseases.

Interest has grown in the consumption of fermented foods as a means to address obesity-related health problems [5,6]. Sake lees, a byproduct of sake brewing, is a fermented food with health benefits [7,8]. Sake lees consist of components of rice, yeast, and koji (a fungus used in the brewing process, the metabolites of which are highly nutritious and rich in amino acids, peptides, and dietary fiber). Animal studies have revealed that sake lees has functional properties and can inhibit cholesterol increase, lower blood pressure, and inhibit liver damage in mice. Moreover, clinical studies in humans have reported improved conditions in the intestinal environmental [9,10]. However, the underlying mechanisms of action of sake lees and the roles of their specific components are unknown.

In this study, we investigated the effect of sake lees on lipid accumulation in preadipocytes using a metabolome analysis. Metabolome analysis is a type of omics analysis that is focused on the metabolites in organisms, and the benefits of a metabolome analysis are that it allows for the discovery of physiological and pathological mechanisms because it comprehensively captures low molecular weight metabolites in the organism [11].

## 2. Materials and Methods

### 2.1. Materials

Sake lees were refermented at 25 °C for 3 days to increase the content of ingredients such as amino acids, resident protein, and dietary fiber. The precipitate from the refermented mixture was collected by filtration and incubated at 60 °C for 2 h to remove the fat. The residue was collected by filtration and was lyophilized and powderized to obtain ISLCs. The composition of ISLCs is shown in Table 1. The ISLCs were diluted in distilled water at a ratio of 9 mL of water to 1 g of ISLCs and were heated at 100 °C for 20 min. The solution was then centrifuged and the supernatant was used for the experiment.

Sake lees, the raw material of ISLC used for this study, has been ingested as a food material in Japan for a long time and is considered a safe food since there is enough food experience in Japan.

**Table 1.** Composition of ISLCs. The content of the components in 100 g of ISLCs are shown.

| Composition | Content (g/100g) |
|---|---|
| Protein | 57.4 |
| Fat | 5.4 |
| Carbohydrate | 29.6 |
| Moisture | 6.5 |
| Ash | 1.1 |
| Dietary fiber | 25.7 |

### 2.2. Cell Culture

Mouse 3T3-L1 cells (Japanese Collection of Research Bioresources Cell Bank, Osaka, Japan) were grown in Dulbecco's modified Eagle's medium (DMEM) (with a high glucose content (4500 mg/L) and antibiotics including 10,000 unit/mL penicillin, 10,000 μg/mL streptomycin, 25 μg/mL amphotericin B, and 10% fetal bovine serum (FBS)) at a temperature of 37 °C in 5% $CO_2$ until confluence was achieved. Cell differentiation was induced by placing the cells in DMEM supplemented with 0.5 mM 3-isobutyl-1-methylxanthine (IBMX), 0.25 μM dexamethasone (DEX), 10% FBS, and 10 μg/mL insulin. The passage number of cells used for the experiments were 11 to 14 passages.

Furthermore, 1.0 mg/mL ISLCs was added to the differentiation medium of the cells to create a group of treated cells, while the remainder of the cells acted as non-ISCLs-treated controls. The cells were incubated for 48 h; the differentiation medium was replaced with DMEM containing 10% FBS and 5 μg/mL of insulin, and this was changed every 48 h for 8 days.

In this study, DMEM prepared was purchased and used for the cell culture and the following experiments. In addition, the cells used were confirmed to be mycoplasma

negative at the time of purchase, so all the experiments were performed in mycoplasma-negative conditions.

### 2.3. Oil Red O Stain

Oil Red O staining was used to assess lipid accumulation in the fully differentiated adipocyte cells collected 8 days after confluence. Adipocytes were fixed with 10% formalin for 1 h, after which the fixing solution was removed. The cells were subsequently stained with filtered 0.3% Oil Red O solution in 60% isopropanol for 1 h before the staining solution was removed by washing with water. Oil Red O was then eluted with isopropanol, and the cells were dried. Absorbance in the eluate was measured at 560 nm using a microplate reader.

### 2.4. Metabolome Analysis

A metabolome analysis was conducted in 3T3-L1 cells collected 4 and 8 days after the initiation of differentiation. After the 3T3-L1 cells were collected, a mixed solvent (MeOH/$H_2O$/CHCl$_3$ = 2.5:1:1) was added, followed by sonication to fracture the cells. A total of 6 µL of 1.0 mg/mL 2-isopropylmalic acid and ribitol as internal standard substances were added and incubated at 37 °C, 1200 rpm for 30 min with shaking and then centrifuged at 4 °C, 19,300× $g$ for 3 min. The supernatant was collected and diluted with Milli-Q and then centrifuged at 4 °C, 19,300× $g$ for 3 minutes. A total of 800 µL of the supernatant was transferred to a new tube, centrifugally concentrated for 2 hours, and lyophilized overnight. To these dried samples, 60 µL of 20 mg/mL methoxyamine containing pyridine solution was added followed by sonication for 20 minutes. The samples were incubated at 30 °C for 90 min at 1200 rpm, and 30 µL MSTFA was then added, followed by incubation at 37 °C for 30 min. The samples were centrifuged at 20 °C for 3 min at 19,300× $g$, and the supernatant was used as the sample. Supernatant samples were analyzed by gas chromatography–mass spectrometry. This method was based on the following literature [12].

### 2.5. Analysis of Free Amino Acids in Cell Culture Medium by High-Performance Liquid Chromatography

The medium in which the cells were cultured was collected at days 4 and 8 after differentiation and filtered through a 0.22 µm filter to prepare the sample for analysis. Valine, leucine, and isoleucine (BCAAs) in the culture medium were analyzed via the ortho-phthalaldehyde fluorescence method by using high-performance liquid chromatography.

### 2.6. Statistical Analysis

All data are expressed as mean ± standard deviation. Dunnett's method was used to assess the lipid accumulation rate, whereas Student's *t*-test was used for the analysis of the metabolome and BCAAs concentrations in the culture medium as determined by high-performance liquid chromatography. Accordingly, $p < 0.05$ and $p < 0.01$ were considered to indicate significant and very significant differences, respectively. Excel Tokei (Version 2.12: Social Survey Research Information Co., Ltd., Tokyo, Japan.) was used for the data analysis.

## 3. Results

### 3.1. Lipid Accumulation

The lipid accumulation rate of each sample of indigestible ingredients was calculated using the fat content of cells in the no-addition group as a control. In the cells treated with 1.0 mg/mL ISLCs, the lipid accumulation was 80.9% ± 6.1% of that in non-ISLCs-treated cells (Figure 1).

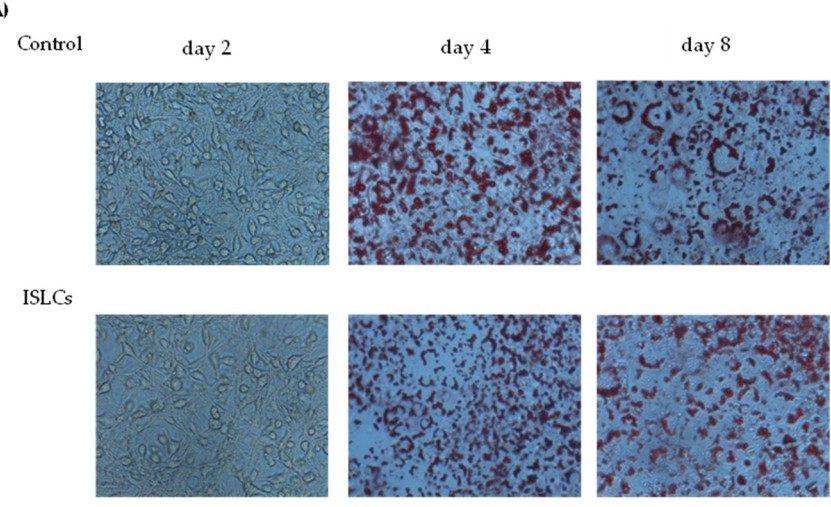

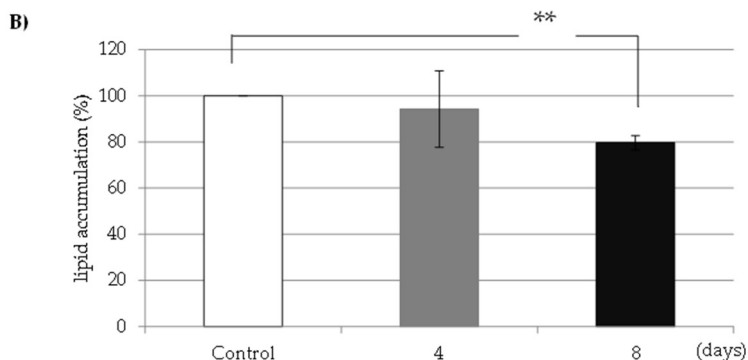

**Figure 1.** ISLCs on fat accumulation in adipocytes. (**A**) Lipid droplets of control cells and ISLCs-treated cells after Oil Red O staining. (**B**) Comparison of lipid accumulation in 3T3-L1 cells. Cell culture was cultivated for 8 days in medium including various kinds of extracts of indigestible sake lees component powder (ISLCs). Fat accumulation was expressed relative to untreated control cells (100%). Values are expressed as mean ± SD; different from control cells; ** $p < 0.01$ (n = 3).

*3.2. Metabolome Analysis*

Metabolites of the adipocytes at days 4 and 8 after differentiation of the control and ISLCs-treated cells were analyzed with metabolomics. As a result, 80 metabolites were detected, and metabolites that were found to be significantly varied between the control and ISLCs-treated cells are shown in Table 1. Compared with the control, 7 and 68 metabolites were found to be significantly increased and 36 and 0 were found to be significantly decreased in the ISLCs-treated cells at 4 and 8 days after differentiation, respectively (Table 2).

The metabolome analysis showed that valine, leucine, and isoleucine were significantly lower in ISLCs-treated cells at day 4 and significantly higher at day 8 after differentiation compared with the control. It is assumed that BCAAs are closely linked to obesity because they function not only as protein constituents but also as factors that regulate various metabolic systems and inhibit insulin resistance.

The variation in the intracellular BCAAs in the control and ISLCs-treated cells at days 4 and 8 after differentiation is shown in Figure 2. In the control cells, there was a significant decrease in the BCAAs from day 4 to day 8 after differentiation (Figure 2a–c), whereas in the ISLCs-treated cells, the BCAAs did not significantly vary from day 4 to 8 after differentiation (Figure 2d–f). Furthermore, the BCAAs levels were higher in the control compared with the ISLCs-treated cells at day 4 after differentiation, whereas they were higher in the ISLCs-treated cells compared with the control at day 8 after differentiation.

**Table 2.** The list of significantly changed metabolites in adipocytes for 4 and 8 days after differentiation. Values are represented as the fold-induction of peak intensity of adipocytes treated by ISLCs and control for 4 and 8 days after differentiation (n = 4). *p* values were calculated according to the Student's *t*-test, andunderlines indicate *p* values lower than 0.05.

| Metabolites | Day 4 | | Day 8 | |
|---|---|---|---|---|
| | *p* Value | Fold Induction | *p* Value | Fold Induction |
| Pyruvic acid | 0.0055 | 0.76 | 0.3270 | 1.11 |
| Lactic acid | 0.0002 | 0.67 | 0.1753 | 1.19 |
| Glycolic acid | 0.2290 | 0.91 | 0.0019 | 1.38 |
| Alanine | 0.1153 | 0.63 | 0.0358 | 3.06 |
| 2-keto-isovaleric acid | 0.8862 | 0.96 | 0.0436 | 2.81 |
| Glycine | 0.8704 | 1.07 | 0.0371 | 2.99 |
| 2-Hydroxybutyric acid | 0.2360 | 0.84 | 0.0003 | 1.84 |
| Sarcosine | 0.4409 | 1.09 | 0.0076 | 1.54 |
| 2-Aminoisobutyric acid | 0.0167 | 1.21 | 0.00002 | 1.51 |
| 3-Hydroxybutyric acid | 0.0025 | 0.69 | 0.0507 | 1.36 |
| 3-Aminopropanoic acid | 0.0080 | 0.26 | 0.1166 | 2.78 |
| Acetoacetic acid | 0.1156 | 0.80 | 0.0169 | 1.69 |
| 3-Hydroxyisovaleric acid | 0.1585 | 0.91 | 0.0152 | 1.56 |
| Valine | 0.0192 | 0.68 | 0.0070 | 1.67 |
| Dihydroxyacetone | 0.0024 | 0.56 | 0.1078 | 1.47 |
| 2-Aminoethanol | 0.1878 | 0.87 | 0.0003 | 1.79 |
| Glycerol | 0.0036 | 0.84 | 0.0021 | 1.38 |
| Leucine | 0.0327 | 0.57 | 0.0225 | 1.60 |
| Octanoic acid | 0.0417 | 1.29 | 0.1766 | 1.32 |
| Phosphoric acid | 0.9007 | 1.01 | 0.0067 | 1.64 |
| Isoleucine | 0.0075 | 0.59 | 0.0043 | 1.92 |
| Proline | 0.0151 | 0.52 | 0.0080 | 2.54 |
| Succinic acid | 0.0011 | 0.56 | 0.0452 | 1.53 |
| Glyceric acid | 0.0019 | 0.58 | 0.5820 | 1.13 |
| Uracil | 0.0023 | 0.73 | 0.0786 | 1.29 |
| Serine | 0.0129 | 0.41 | 0.0022 | 2.39 |
| Fumaric acid | 0.5168 | 0.95 | 0.0037 | 1.65 |
| Homoserine | 0.1694 | 1.14 | 0.0272 | 1.95 |
| Nonanoic acid | 0.0633 | 1.14 | 0.0063 | 1.47 |
| Threonine | 0.0232 | 0.63 | 0.0010 | 2.73 |
| Glutaric acid | 0.0992 | 0.77 | 0.0347 | 1.74 |
| Citramalic acid | 0.1531 | 0.89 | 0.0114 | 2.03 |
| Threitol | 0.0383 | 0.79 | 0.0498 | 1.83 |
| Malic acid | 0.0428 | 0.78 | 0.0492 | 1.83 |
| meso-Erythritol | 0.0023 | 0.81 | 0.0374 | 1.84 |
| Aspartic acid | 0.0231 | 0.56 | 0.0001 | 2.04 |
| 3-Aminoglutaric acid | 0.0252 | 0.56 | 0.0001 | 2.03 |
| 4-Hydroxyproline | 0.0102 | 0.61 | 0.0128 | 2.51 |
| Methionine | 0.0153 | 0.59 | 0.0027 | 2.31 |
| 4-Aminobutyric acid | 0.0670 | 1.15 | 0.0056 | 2.75 |
| 5-Oxoproline | 0.1823 | 0.88 | 0.0206 | 2.30 |
| Cysteine | 0.0100 | 0.53 | 0.0011 | 2.18 |
| Creatinine | 0.1297 | 0.78 | 0.0026 | 1.61 |
| 2-ketoglutaric acid | 0.7964 | 1.04 | 0.0071 | 1.99 |
| Glutamic acid | 0.0148 | 0.55 | 0.0041 | 1.75 |
| 5-Aminovaleric acid | 0.3171 | 1.18 | 0.0010 | 2.03 |
| Xylose | 0.0136 | 0.85 | 0.0268 | 2.05 |
| Phenylalanine | 0.0030 | 0.63 | 0.0077 | 2.33 |
| Lyxose | 0.8255 | 0.98 | 0.0004 | 1.46 |
| Arabinose | 0.0074 | 0.80 | 0.0164 | 1.84 |

**Table 2.** *Cont.*

| Metabolites | Day 4 | | Day 8 | |
|---|---|---|---|---|
| | *p* Value | Fold Induction | *p* Value | Fold Induction |
| Ribulose | 0.6849 | 0.93 | 0.0118 | 1.39 |
| Lauric acid | 0.0909 | 1.08 | 0.0062 | 1.41 |
| Ribose | 0.5429 | 1.05 | 0.0007 | 1.48 |
| Homocysteine | 0.0041 | 0.63 | 0.0285 | 2.02 |
| Asparagine | 0.0864 | 0.65 | 0.0045 | 2.04 |
| Xylitol | 0.0292 | 0.72 | 0.0095 | 1.85 |
| Arabitol | 0.2740 | 1.04 | 0.0001 | 1.42 |
| 2-Deoxy-glucose | 0.5964 | 1.03 | 0.00004 | 1.42 |
| Orotic acid | 0.0001 | 0.72 | 0.0005 | 1.74 |
| Isocitric acid | 0.0440 | 1.40 | 0.0623 | 2.95 |
| 2-Aminopimelic acid | 0.0256 | 1.50 | 0.0350 | 3.09 |
| Citric acid | 0.0306 | 1.50 | 0.0343 | 3.16 |
| Ornithine | 0.7128 | 0.88 | 0.0249 | 1.84 |
| Glycyl-Glycine | 0.1939 | 0.90 | 0.0045 | 1.54 |
| Cadaverine | 0.0463 | 0.84 | 0.0039 | 1.47 |
| Mannose | 0.0048 | 0.73 | 0.0174 | 1.78 |
| Glucose | 0.0015 | 0.47 | 0.3026 | 2.10 |
| Galacturonic acid | 0.1774 | 0.87 | 0.0058 | 1.84 |
| Glucuronic acid | 0.1764 | 0.87 | 0.0069 | 1.84 |
| Tryptamine | 0.0584 | 0.82 | 0.0219 | 1.85 |
| Glucaric acid | 0.0272 | 0.72 | 0.0077 | 1.71 |
| Palmitoleic acid | 0.0024 | 2.13 | 0.0508 | 1.41 |
| Inositol | 0.2936 | 0.91 | 0.0014 | 1.43 |
| Margaric acid | 0.0337 | 1.14 | 0.0023 | 1.38 |
| Kynurenine | 0.0583 | 0.73 | 0.0002 | 1.54 |
| 2,3-bisphosphoglyceric acid | 0.0936 | 0.77 | 0.0005 | 1.58 |
| Cystamine | 0.0408 | 0.78 | 0.0091 | 1.43 |
| Elaidic acid | 0.9684 | 0.98 | 0.0328 | 1.29 |
| Stearic acid | 0.9072 | 1.01 | 0.0042 | 1.37 |
| Maltose | 0.0141 | 0.78 | 0.1247 | 1.67 |

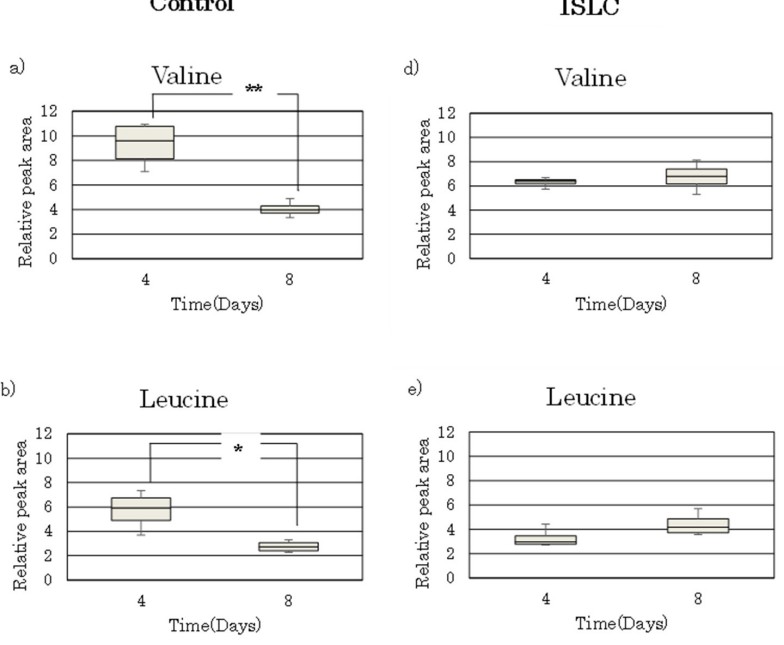

**Figure 2.** *Cont.*

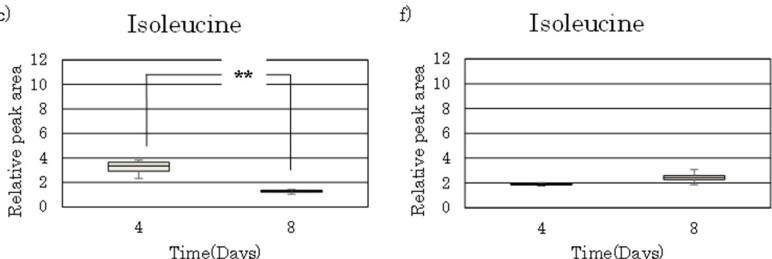

**Figure 2.** Comparison of BCAAs in adipocytes for 4 and 8 days after differentiation. We performed the metabolome analysis for adipocytes of 2 different time points (day 4 and 8 of adipogenesis). Metabolite relative peak areas are shown as box plots. Alteration patterns of BCAAs are shown for control cells (**a–c**) and cells treated with ISLCs (**d–f**). *p* values were calculated according to the Student's *t*-test; * *p* < 0.05, ** *p* < 0.01 (n = 4).

### 3.3. Analysis of BCAAs in Cell Culture Medium

The BCAAs in the medium of the control and ISLCs-treated cells at days 0, 4, and 8 after differentiation were analyzed with high-performance liquid chromatography, and the results are shown in Figure 3. In both the control (Figure 3a–c) and ISLCs-treated cells (Figure 3d–f), the BCAAs levels in the medium were high on day 0 and decreased significantly at day 4 after differentiation. The values at day 8 were almost same as at day 4.

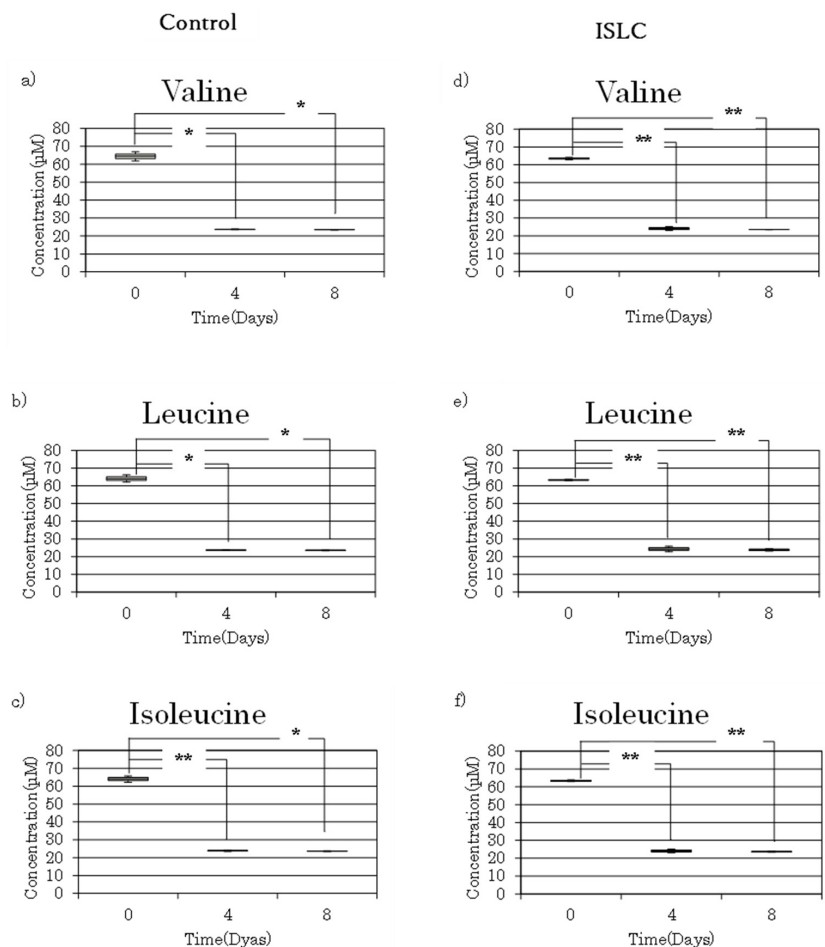

**Figure 3.** Comparison of BCAAs in medium for 4 and 8 days after differentiation. We measured concentrations of BCAA in medium of 3 different time points (day 0, 4, and 8 of adipogenesis) by HPLC. Alteration patterns of BCAA are shown for control cells (**a–c**) and cells treated with ISLCs (**d–f**). *p* values were calculated according to the Student's *t*-test: * *p* < 0.05, ** *p* < 0.01.

## 4. Discussion

In this study, we investigated the effects of ISLCs on lipid accumulation in 3T3-L1 adipocytes. The results revealed that the addition of 1.0 mg/mL ISLCs to the culture medium was associated with significantly less lipid accumulation compared to the untreated controls; thus, ISLCs may contain components that inhibit lipid accumulation in adipocytes (Figure 1). The specific functional component that has the effect of inhibiting fat accumulation in ISLCs is currently under investigation, but since the effect was caused by the sample extracted at a high temperature at 100 °C, we consider that the functional component may be a component that is extracted at a high temperature at 100 °C.

Furthermore, the results from the metabolome analysis of the adipocytes at 4 and 8 days after differentiation showed differences in certain metabolites between the control and ISLCs-treated cells, especially in the amino acids (Figure 2).

Metabolomic studies on preadipocytes and mature adipocytes have shown that BCAAs metabolism is strongly regulated during adipogenesis and in mature adipocytes, which indicates that BCAAs are involved in adipocyte differentiation and maturation as well as lipid accumulation [13].

In this study, BCAAs levels in the control cells were higher at day 4 compared with day 8 after differentiation (Figure 2a–c), and BCAAs levels in the control medium decreased from day 0 to 4 after differentiation (Figure 2d–f). The adipocytes at day 4 after differentiation were in the preparation stage for differentiation from preadipocytes to adipocytes, and the BCAAs required for differentiation are thought to be stored in cells from the culture medium, and we considered this to be the reason for the high intracellular BCAAs concentrations. Next, the BCAAs levels in the cells decreased from day 4 to 8 after differentiation (Figure 2a–c), whereas they did not change in the medium (Figure 3a–c). This fact suggests that the intracellular BCAAs levels were reduced at day 8 after differentiation because the BCAAs required for adipose synthesis were incorporated back into the cells from the culture medium and then utilized for the adipose tissue, which allows the preadipocytes to differentiate into adipocytes and lipids to accumulate. According to the metabolome analysis of the cultured adipocytes by Halama et al., the BCAAs leucine is metabolized from day 4 after differentiation to synthesize cholesterol, and isoleucine contributes to the addition of a fatty acid chain after metabolism, indicating that the two amino acids are deeply involved in lipid synthesis [13]. In this study, leucine and isoleucine in control cells were metabolized and decreased from day 4 to 8 after differentiation, which suggests that leucine and isoleucine were involved in lipid synthesis.

The number of adipocytes is thought to be established in childhood and adolescence, and the number of adipocytes is constant in adults, independent of weight gain or loss and nutritional status [14]. Furthermore, it has been found that obese children and young adults have higher BCAAs concentrations in their cells compared with healthy individuals, whereas obese adults have lower BCAAs concentrations in their cells [15,16].

In this study, the BCAAs concentration was high in the control cells at day 4 after differentiation, which was the differentiation stage, which suggests that the cells at day 4 after differentiation reflected the state of adipocytes in childhood and adolescence, and the BCAAs concentration was low in the control cells at day 8 after differentiation, which is the maturation stage, which suggests that the cells at day 8 after differentiation reflected the state of adipocytes in adults.

On the other hand, the BCAAs in the cultured medium of the ISLCs-treated cells were found to decrease from day 0 to day 4 after differentiation (Figure 3a–c). This may have been due to the fact that the preadipocytes incorporated the BCAAs necessary for their differentiation into the adipocytes from the culture medium and into the cells, which also occurred with the control. However, the BCAAs concentrations in the ISLCs-treated cells were not significantly different between day 4 and 8 after differentiation, which was different from the control (Figure 3d–f). Amino acids are not only involved in protein synthesis but also have an important role in reducing surplus body fat and inhibiting obesity caused by diet. In a mouse experiment, it was confirmed that leucine improved

glucose and cholesterol metabolism and inhibited obesity [17], and isoleucine intake increased the expression of UCP, PPARα, and CD36, and inhibited lipid accumulation and hyperglycemia [18].

Furthermore, regarding the non-BCAAs, it has been suggested that threonine intake controls the expression of genes such as UCP and inhibits weight gain, lipid accumulation, and adipocytes growth in mice [19]; additionally, it has also been found that glutamine intake inhibits weight gain and improves hyperglycemia and hyperinsulinemia status, as seen in mouse experiments [20]. Details of the mechanism of the inhibitory effects of these amino acids on lipid synthesis and obesity have not been elucidated, but it is thought that the improvement in insulin resistance and increased oxidation of fatty acids caused increased energy consumption [20].

The metabolome analysis conducted in this study confirmed that higher levels of BCAAs, threonine, and glutamine, which have been reported to inhibit lipid accumulation and obesity in previous studies, were observed in ISLCs-treated cells compared with control cells (Table 2). Therefore, we believe that ISLCs are involved in the metabolism of amino acids in adipocytes and that they regulate the expression genes related to lipid synthesis, such as UCP and PPARα, which inhibit lipid accumulation. However, gene expression was not investigated in this study, and therefore, such analysis will be of interest in future research that continues from this study.

In summary, the findings of this study suggest that the intake of sake lees inhibits obesity and can be potentially applied in the reduction in obesity and metabolic syndrome, which is a global issue.

**Author Contributions:** All authors contributed to writing and editing this paper (Y.M., S.N., M.Y. and M.T.). All authors have read and agreed to the published version of the manuscript.

**Funding:** This research was supported in part by Kobe College and Kobe University.

**Institutional Review Board Statement:** Not applicable.

**Informed Consent Statement:** Not applicable.

**Data Availability Statement:** Data is contained within the article.

**Conflicts of Interest:** The authors declare no conflict of interest.

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
