# Peer review of "Metabolome Analysis of the Effects of Sake Lees on Adipocyte Differentiation and Lipid Accumulation"

_fermentation, doi:10.3390/fermentation9030300_

Round 1

Reviewer 1 Report

Minor comments.

Separate unit and concentration 1 mg/mL

Why did you referment the sake lees? The sake lees fermented at 25°C for 3 days.

What antibiotics do you add?

CO2

“Furthermore, 1.0 mg/mL ISLCs”, Do you use the supernatant of the ISLCs or the ISLCs directly??

1200 rpm and 16,000 rpm better in x g

“gas chromatography-mass spectrometry”, Do you have a reference for the chromatographic conditions?

“Analysis of free amino acids”, Do you have a reference for the chromatographic conditions?

I don't understand how you calculate the fold induction, can you explain it? It is the logarithm of the ratio of sample to control, another type of calculation?

“We found insoluble fractions with a molecular weight under 10,000”, Is it the result of previous work? Otherwise where to find this result in your manuscript? This insoluble fraction corresponds to protein, polysaccharide, both, other chosen??

Author Response

Response to Reviewer 1 Comments

Point 1: Separate unit and concentration 1 mg/mL.

Response 1: For all 1.0 mg/ml, units and concentrations are separated.

Point 2: Why did you referment the sake lees? The sake lees fermented at 25°C for 3 days.

Response 2: The reason for refermenting the sake lees is to increase the content of the ISLCs' major components, resistant protein and dietary fiber. We added this content to lines 65-66.

Point 3: What antibiotics do you add?

Response 3: For antibiotics, Gibco Antibiotic-Antimycotic solution containing penicillin, streptomycin, and Amphotericin B was used to prevent cell culture contamination. We added antibiotics information to lines 77-78.

Point 4: “Furthermore, 1.0 mg/mL ISLCs”, Do you use the supernatant of the ISLCs or the ISLCs directly??

Response 4: We used the supernatant. (line 71)

Point 5: 1200 rpm and 16,000 rpm better in x g “gas chromatography-mass spectrometry”, Do you have a reference for the chromatographic conditions?

Response 5: In this experiment, 1200 rpm is not centrifugation and cannot be changed, but 16,000 rpm can be changed to 19,300 x g. We referred to the following paper for metabolome analysis in this study. We added this paper to refernces [12].

Yoshie T., Nishiumi S., Izumi Y, Sakai A., Inoue J., Azuma T., Yoshida M. (2012) Regulation of the metabolite profile by an APC gene mutation in colorectal cancer. Cancer Science, 103, 1010-1021.

Point 6: “Analysis of free amino acids”, Do you have a reference for the chromatographic conditions?

Response 6: The analysis of free amino acids in this study was performed with reference to the following paper for amino acid analysis in this study.

Hitachi High-Tech, LC Application Sheet AS/LC-010

Point 7: I don't understand how you calculate the fold induction, can you explain it? It is the logarithm of the ratio of sample to control, another type of calculation?

Response 7: "fold induction" is the ratio of sample to control relative peak area.

Point 8: “We found insoluble fractions with a molecular weight under 10,000”, Is it the result of previous work? Otherwise where to find this result in your manuscript? This insoluble fraction corresponds to protein, polysaccharide, both, other chosen?

Response 8: The "insoluble fraction with a molecular weight of 10,000 or less" is currently under experimentation and has not yet been published. The specific composition has not yet been identified and will be corrected in the future.

Reviewer 2 Report

1) Whether cell cultures have been tested for mycoplasma contamination. This fact should be reflected in materials and methods.

2) Information about the passage from which the cells were used for experiments should be displayed.

Author Response

Response to Reviewer 2 Comments

Point 1: Whether cell cultures have been tested for mycoplasma contamination. This fact should be reflected in materials and methods..

 Response 1: In this study, DMEM prepared was purchased and used for cell culture and the following experiments. In addtion, the cells used have been confirmed to be mycoplasma-negative at the time of purchase, so all the experiments must be perfomed at the mycoplasma-negative condition. We added this content to lines 89-92.

Point 2: Information about the passage from which the cells were used for experiments should be displayed.

Response 2: The passage numbers of cells used for the experiments were 11 to 14 passages. We added this content to lines 82-83.